# Thifluzamide, Fludioxonil, and Clothianidin as Seed Treatment Can Efficiently Control Major Soil-Borne Diseases, Aphids (*Aphidoidea* spp.), and Residue Distribution in the Field

Chao Chen [1], Xumiao Wang [1], Shanshan Yin [1], Chao Wang [1], Xuexiang Ren [2], Quan Gao [1] and Haiqun Cao [1,*]

1   Anhui Province Engineering Laboratory for Green Pesticide Development and Application, School of Plant Protection, Anhui Agricultural University, Hefei 230036, China
2   Institude of Plant Protection and Agro-Products Safety, Anhui Academy of Agricultural Science, Hefei 230001, China
*   Correspondence: haiquncao@163.com

**Abstract:** Combined seed treatment with neonicotinoids and fungicides offers a potential control measure for pest management at the wheat seeding stage. In this study, a novel, highly-efficient seed-coating agent was prepared using thifluzamide, fludioxonil, and clothianidin as its active components and other additives (abbreviated to TFC). Laboratory experiments and field trials revealed a positive effect on germination, plant height, and root length, with 90% control efficiency on wheat sharp eyespots and aphid infestations. Meanwhile, the distribution of thifluzamide, fludioxonil, and clothianidin residues in the wheat plants at harvest was below 0.05 mg/kg both at the recommended dosage and at 2.0 times the recommended dose. Furthermore, an artificial soil assay of biotoxicity in earthworms revealed a low level of toxicity at $LC_{50} > 10$ mg/kg. Overall, these findings suggest that TFC has the potential to control major soil-borne diseases and pest infestations in wheat, offering an environmentally-friendly alternative to more toxic pesticides.

**Keywords:** seed treatment; wheat sharp eyespot; aphids; earthworms; pesticide



## 1. Introduction

Wheat (*Triticum aestivum* L.) is one of the most important food crops in the world, thereby playing a significant role in global food security [1–3]. However, soil-borne diseases and pests such as wheat sharp eyespots (*Rhizoctonia cerealis*) and aphids (*Aphidoidea* spp.) represent serious problems in wheat production [4–6]. At present, chemical foliar sprays are the main control measure employed in wheat fields in China [7]. However, due to the broad host ranges and difficulties in targeting pathogen populations in the soil, these sprays tend to be inefficient [8]. Meanwhile, the high fecundity of aphids and pest resistance to commonly used insecticides have further reduced the overall efficiency, as well as increased the associated labor costs [9,10].

The seed coating agent is a kind of pesticidal preparation that could be used to treat the seeds with the tendency to form a film coat [1]. Seed coating agents are generally composed of active constituents (pesticide and plant growth regulators) and inactive components, including film-forming agents, suspension concentrates, and pigments. These are important measures in integrated disease and pest management and are currently used in agricultural areas worldwide [11]. These treatments also represent an alternative to foliar spray, providing longer-term protection as well as reducing environmental toxicity [12,13]. Studies have shown that seed coating is effective in preventing and controlling diseases and pests while promoting seedling growth and increasing yield, thereby offering an environmentally-safe, easy-to-use alternative to toxic pesticide use [14,15]. In China, fungicide seed treatments are currently widely used to control wheat seedling diseases such as *Rhizoctonia cerealis* and *Bipolaris sorokiniana* [16]. Furthermore, neonicotinoid seed treatments are also used

to control a wide range of agricultural pests, especially sucking-piercing insects such as whitefly, thrips [17], and cotton aphids [18,19], providing early-season protection of wheat seedlings in the field [20,21]. However, single-seed coating agents tend to lack efficiency and, in addition, often contain large amounts of toxic pesticides such as carbofuran and imidacloprid, which pose a risk to both the environment and human health [22,23]. Meanwhile, few reports have documented the overall safety of seed coating agents.

Numerous studies have suggested that neonicotinoid seed treatment combined with fungicide application could act as a potential control measure for pest management at the wheat seeding stage [24]. In this regard, we successfully prepared an environmentally-friendly complex wheat seed coating agent consisting of thifluzamide, fludioxonil, and clothianidin (the 10% thifluzamide · fludioxonil · clothianidin, short for TFC) as an alternative to more traditional toxic products. In this study, laboratory and field experiments were carried out to compare the control efficiency of TFC and the conventional seed coating agent (the 27% difenoconazole · fludioxonil · thiamethoxam, short for DFT) on wheat sharp eyespots and aphid infestations. The effect on seed germination and growth was also studied. In addition, the concentration of thifluzamide, fludioxonil, and clothianidin residues in the wheat plants and soil was also determined, while a biotoxicity assay was carried out on earthworms. The results provide a foundation for the rational use of TFC as an alternative to more environmentally-damaging pesticide controls.

## 2. Materials and Methods

### 2.1. Chemicals and Reagents

High-purity pesticide standards of thifluzamide (99.2% purity), fludioxonil (99.5% purity), clothianidin (99.9% purity), imidacloprid (99.1% purity), nitenpyram (98.2% purity), thiacloprid (99.2% purity), acetamiprid (97.2% purity), and thiamethoxam (98.2% purity) were purchased from Suolaibao Laboratory Technologies Inc. (Beijing, China). Adsorbent primary secondary amine (PSA; 500 mg, 6 mL), analytical reagent grade anhydrous magnesium sulfate ($MgSO_4$), and anhydrous sodium chloride (NaCl) were obtained from Maikelin Technologies (Shanghai, China). The water used was purified with a Milli-Q Gradient A10 system (Millipore, Germany). All the solvents and remaining chemicals were of chromatographic grade and used without further purification. Standard stock solutions of thifluzamide, fludioxonil, and clothianidin (1000 mg/L) were prepared individually in acetonitrile and stored at $-20\,°C$.

### 2.2. Experimental Conditions

The wheat seeds used in this study were Yangmai 20, one of the most popular winter wheat cultivars in China, which is moderately resistant to comprehensive disease. Pathogen strains *R. cerealis* and *B. sorokiniana* were isolated, and a sensitive strain of uniformly-sized wingless wheat aphids was provided by the Anhui Province Engineering Laboratory for Green Pesticide Development and Application, the School of Plant Protection, and the Anhui Agricultural University. The fungal strains were cultivated in potato dextrose agar (PDA) at 25 °C in the dark and then stored on PDA slants at 4 °C.

### 2.3. Preparation of the TFC Seed Coating Agent

The selection of fungicides was carried out in vitro using the above two soil-borne diseases based on the hyphal growth rate method. Based on the half-maximal effective concentrations ($EC_{50}$), thifluzamide and fludioxonil were prepared at ratios of 10:1, 5:1, 2:1, 1:1, 1:2, 1:5, and 1:10. PDA plates were then, respectively, inoculated with *R. cerealis* and *B. sorokiniana* blocks with a diameter of 5 mm. Tests were replicated three times at 25 °C for 7 d. After incubating for 3 days, the diameter of the colony (mm) on the PDA was then measured using a cross-measuring method, and the mycelial growth inhibition rate (%) of

each treatment was calculated according to the following formula, and the $EC_{50}$ value was calculated using SPSS 17.0 software.

$$Inhibitory\ rate\ (\%) = \frac{Diameter\ of\ control - Diameter\ of\ treatment}{Diameter\ of\ control - 0.6\ cm} \times 100$$

The synergistic effect was also determined using the co-toxicity coefficient (CTC), as described previously [25]. Briefly, $EC_{50}$ values were determined using the growth rate method and then converted into the actual toxicity index (ATI) and the theoretical toxicity index (TTI). Values were then used to calculate the CTC as follows:

$$CTC = \frac{ATI}{TTI} \times 100$$

where a value greater than 120 indicates synergism, a value between 80 and 120 indicates an additive effect, and a value less than 80 indicates antagonism.

The leaf-dip method was subsequently used to determine the activity of different concentrations of six neonicotinoids against wheat aphids [26,27]. Briefly, 0.1, 0.2, 0.5, 1, and 2 mg/L of each neonicotinoid was prepared by dissolving in DMSO with 0.01% tween 80 and then diluting to the desired concentration with water. Wheat leaves were then dipped in the solution for five seconds and allowed to dry for 45 min at room temperature. Each treatment was repeated three times with 15 larvae per replicate and control. Samples were then placed in a growth chamber at a constant temperature of 25 °C. Larval mortality was then recorded 48 h after treatment, and the median lethal concentration ($LC_{50}$) was calculated from Probit analysis with 95% confidence limits.

The seed coating agent was prepared by the wet sand processing superfine grinding method [28]. The optimal formula for the seed coating agent was determined using an orthogonal test. The procedure conditions were as follows: all the ingredients, such as thifluzamide, fludioxonil, clothianidin, NNO, LAE-9, S-20, XG, magnesium aluminum silicate, and pigment, were mixed according to a certain proportion to obtain the aqueous solution of the desired consistency. The water and surfactant ratios were selected on the basis of previous studies. A rotor-stator homogenizer was used to make the pesticide active ingredient form a stable dispersion system [28]. After mixing thoroughly at room temperature, the mixture was placed on an electron constant speed mixer (2000 rpm) for 2–3 h until completely dissolved. Then, other additives (including the film-forming auxiliaries, etc.) were added to the aqueous solution according to a certain ratio to obtain a novel seed coating agent. Meanwhile, the stability was measured by the recommended Collaborative International Pesticides Analytical Council (CIPAC) method, which well fulfilled the demands of pesticide preparation.

### 2.4. Laboratory Experiments

Wheat seeds were coated with TFC and then air-dried for 20 min. Five coating treatments were then examined in the laboratory: uncoated wheat seeds as a blank control (CK) and seeds coated with TFC by hand at ratios of 1:50, 1:100, 1:200, and 1:500 (agent: seed). According to the guidelines of the International Seed Testing Association (ISTA) [29], 100 seeds were then sampled per group and placed evenly in a pot containing soil. They were then incubated in an incubator at 20 ± 1 °C with 85% relative humidity. Irrigation was carried out as required every two days. Each pot was replicated three times and then observed for seven days for signs of germination. On days three and seven, the germination potential and germination percentage were calculated, respectively. A random sample of 15 seedlings from each treatment was then selected to determine seedling quality. The optimal formulation developed in the laboratory was then selected based on its overall performance and used for evaluation in the field.

A pot experiment was also conducted to determine the control effect of TFC on wheat sharp eyespot disease. Treatments were conducted as in the above germination test. Briefly, a PDA medium containing *R. cerealis* was mixed into the soil of three pots per treatment.

The percentage of plants showing browning symptoms on the base of the stem was then calculated as previously described [30].

A pot experiment was also performed to determine the effect of TFC on aphid infestation. Leaves of wheat plants at the one-leaf stage and one core were inoculated with adult wheat aphids. The next day, the number of aphids on the leaves was then recorded. Each pot was replicated three times, with 20 aphids per replicate.

### 2.5. Field Experiments

In addition, laboratory experiments and field trials were also carried out to compare the effects of TFC and the traditional coating agent DFT (purchased from Syngenta company, Basel, Switzerland). Dissipation and final residue analyses were carried out in accordance with the Standard Operating Procedures on Pesticide Registration Residue Field Trials (NY/T 788-2018) issued by the Institute for the Control of Agrochemicals, Ministry of Agriculture, China. Trials were conducted in 2021–2022 in the wheat experimental plots of the Anhui Academy of Agricultural Sciences, China. Wheat seeds were sown according to standard agronomic practices in late October, and seed treatment was carried out as described above for the laboratory experiments. Three coating treatments were designed in a randomized complete block design with three replicates each based on the results of the above analyses: the untreated control treatment (CK); TFC at an agent: seed ratio of 1:100; and DFT at the recommended dosage of 1:300. Each plot consisted of 10 five-meter-long rows, separated by 1.5 m of bare, cultivated ground. A random sample of 100 plants from each plot was selected to determine the root and shoot lengths and fresh and dry weights under each treatment. After the heading stage, an additional 100 plants from each plot were randomly sampled to evaluate the control effect on wheat sharp eyespot. For each treatment, the incidence was recorded as the percentage of plants showing browning symptoms on the base of the stem. The reference for the control effect calculation method is Koycu et al. 2019. When the wheat sharp eyespot was identified, the number of aphids was also monitored until it decreased to a very low density.

A residue experiment was also performed at two dosages of TFC: coating with TFC at a mass ratio of 1:100 (agent:seed)(the recommended dosage) and 1:50 (two times the recommended dosage). Each experimental plot covered an area of 30–50 m$^2$ and was replicated three times. The soil surrounding the plants was sampled using a soil auger (0–10 cm) for 2 h, then 3, 14, 21, 30, 54, 73, 124, and 135 days after wheat sowing. Plant samples were also collected 23, 25, 27, 34, 41, 65, 74, 124, and 135 days after sowing for analysis of TFC residues. Soil samples were sieved to remove dirt and gravel and then pooled into approximately 2 kg samples, placed in polythene bags, tagged, and stored at −20 °C until use. Grain samples were collected at harvest, pooled into approximately 2 kg samples, and then grounded using a blender and frozen at −20 °C until use.

### 2.6. Analysis of Pesticide Residues in the Soil and Plant Samples

For the extraction of thifluzamide, fludioxonil, and clothianidin residues, the QuEChERS method was used based on previous validation of its effectiveness. For the analysis of thifluzamide, homogenized soil, plant, and wheat grain samples (5 g $\pm$ 0.01 g each) were, respectively, weighed and then placed in 50 mL centrifugation tubes. Ultrapure water (10 mL), followed by 20 mL of acetonitrile, was then added to each tube and shaken for 60 min at 300 rpm using a mechanical shaker. Anhydrous NaCl (4 g) was then added to the homogenized sample to help separate the acetonitrile from the water before centrifuging at 4000 rpm for 5 min. A total of 6.00 mL of supernatant extract was then filtered into 15 mL centrifuge tubes containing 0.10 $\pm$ 0.01 g PSA, 0.20 $\pm$ 0.01 g anhydrous MgSO$_4$, and 0.10 $\pm$ 0.01 g C$_{18}$. The samples were then vortexed for 1 min and centrifuged at 5000 rpm for 5 min. Aliquots (1 mL) of the extract were then filtered through a 0.22 μm membrane filter and injected into an ultra-performance liquid chromatography-tandem mass spectrometer (UPLC-MS/MS; Waters, Milford, MA, USA) for analysis.

Meanwhile, for the analysis of fludioxonil and clothianidin, chopped and blended representative soil, plant, and grain samples (5 g each) were, respectively, weighed and placed in 50 mL polypropylene centrifuge tubes, and mixed with 10.00 mL acetonitrile and 3.00 mL distilled water. NaCl (1 g) and anhydrous $MgSO_4$ (3 g) were then added to the homogenized samples for phase separation followed by vigorous manual shaking for 1 min. This was followed by centrifugation at 4000 rpm for 5 min. Top layer samples (2 mL) were then transferred to 15 mL centrifuge tubes containing $0.05 \pm 0.01$ g PSA, $0.10 \pm 0.01$ g anhydrous $MgSO_4$, and $0.05 \pm 0.01$ g $C_{18}$. This was followed by centrifugation at 5000 rpm for 5 min. Samples of supernatant extract (1 mL) were then collected and evaporated to dryness under a stream of nitrogen at 40 °C before adding acetonitrile to a final volume of 1 mL. The samples were then passed through a 0.22 µm membrane filter and transferred to UPLC-MS/MS sample vials for analysis.

For UPLC-MS/MS, the column used was a UPLC BEH $C_{18}$ column ($2.1 \times 50$ mm, 1.7 µm; Waters) at 40 °C. The mobile phase consisted of water containing 0.1% (*v/v*) formic acid and acetonitrile at a ratio of 20:80, at an injection volume of 5 µL, and at a flow rate of 0.2 mL min$^{-1}$. The total running time was 5 min. The retention times of pesticides in their respective mobile phases were 2.23 (thifluzamide), 1.35 (fludioxonil), and 0.78 min (clothianidin). Tandem mass spectrometry was performed in the positive electrospray ionization (ESI+) mode, with the following source parameters: gas temperature, 500 °C; desolvation gas flow, 1000 L/h; capillary voltage, 1.60 kV; and argon gas as the collision gas. Detection was carried out in the multiple reaction monitoring (MRM) mode using electrospray positive ionization. The MRM parameters are detailed in Table S2, including the precursor ion, production, fragmentation voltage, and collision energies.

To determine the quantities of thifluzamide, fludioxonil, and clothianidin residues in the soil, plant, and grain samples, retention times and peak areas were compared against standard calibration curves. To avoid deviation due to matrix effects, matrix-matched standards were followed. Recovery studies were also conducted to establish the reliability of the given method and to assess the efficiency of extraction and cleanup. Samples in multiple replicates were, respectively, spiked with different concentrations of each compound to determine the limit of quantification (LOQ) and limit of detection (LOD) with satisfactory relative standard deviations (RSDs).

*2.7. Biosafety of TFC in Earthworms*

The biosafety of TFC was also examined using earthworms using the lethal test method based on pesticide registration guidelines (GB/T 31270.15-2014) [31]. Artificial soil was used to simulate the preferred habitat of earthworms and to provide relatively accurate results for the toxicity assay. The artificial soil comprised 10% sphagnum peat moss, 20% kaolinite clay, 68% quartz sand, and 2% calcium carbonate with a pH of 6.41 [32]. Mature earthworms (*Eisenia foetida*) with a well-developed clitellum were obtained from the YiLong Earthworm Farm in Jurong, Jiangsu, China. Worms with an average weight of 0.4322 g were then selected for the toxicity test. All test devices containing the worms were placed in a climate room at $20 \pm 1$ °C under $80 \pm 5$% relative humidity (see Table S1).

Based on pre-test results, TFC was mixed into the soil as an aqueous solution at the following concentrations (mg/kg, dry soil): 6.58, 9.87, 14.8, 22.2, 33.3, and 50 mg/kg. The soil samples were then transferred into glass jars. Ten adult earthworms were then placed in 1 L glass containers filled with 500 g of soil and enclosed in a polythene sheet consisting of integrated gauze to ensure optimal ventilation. After 7 and 14 days, toxic symptoms were observed in the earthworms. Briefly, living worms were sorted by hand, then the median lethal concentration (LC$_{50}$) and associated 95% confidence limit were calculated [33]. Three replicates were conducted per toxicity test. Solvent and control treatments were also conducted with three replicates each. At the end of the test, the mortality rate of the control treatment was expected to be no more than 10%.

*2.8. Data Analyses*

All data are expressed as the means of three replicated treatments, with standard deviations (SD). SPSS 17.0 was used to determine statistical significance with one-way ANOVA. $LC_{50}$ and associated 95% confidence limits were calculated using SPSS 17.0 software, and *p* values of 0.05 were used to define statistical significance. Figures were drawn using GraphPad Prism 8.

## 3. Results and Discussion

*3.1. Preparation of the TFC Seed Coating Agent*

Thifluzamide is a highly efficient, broad-spectrum systemic fungicide used widely within China to control wheat sharp eyespots. As shown in Table 1, thifluzamide showed significant antifungal activity against *R. cerealis* with an $EC_{50}$ of 0.0189 mg $L^{-1}$ after 7 days. Similarly, fludioxonil was previously found to show obvious inhibitory activity against *B. sorokiniana* ($EC_{50}$ = 0.0101 mg $L^{-1}$) [29]; therefore, the synergistic effect of these two fungicides on soil-borne diseases was examined. Accordingly, the results showed a significant synergistic effect against the test fungi. At a mass ratio of 2:1 (thifluzamide: fludioxonil), the $EC_{50}$ values were 0.0210 and 0.0220 mg $L^{-1}$ against *R. cerealis* and *B. sorokiniana*, respectively. In addition, CTC values of 117.89 and 137.73 were obtained, highlighting the significant potential of this synergistic combination in the control of soil-borne diseases in wheat.

**Table 1.** Experimental design of the fungicide co-formulations and their effectiveness against *R. cerealis* and *B. sorokiniana*.

| Fungicide | Mass Ratio | *R. Cerealis* | | *B. Sorokiniana* | |
|---|---|---|---|---|---|
| | | $EC_{50}$(mg $L^{-1}$) | CTC | $EC_{50}$ (mg $L^{-1}$) | CTC |
| Thifluzamide | - | 0.0189 | - | 1346.7960 | - |
| Fludioxonil | - | 0.0651 | - | 0.0101 | - |
| Thifluzamide: Fludioxonil | 10:1 | 0.0240 | 84.18 | 0.1550 | 71.67 |
| | 5:1 | 0.0210 | 102.07 | 0.0600 | 101.00 |
| | 2:1 | 0.0210 | 117.89 | 0.0220 | 137.73 |
| | 1:1 | 0.0270 | 108.50 | 0.0330 | 61.21 |
| | 1:2 | 0.0320 | 112.10 | 0.0190 | 79.74 |
| | 1:5 | 0.0440 | 105.13 | 0.0230 | 52.70 |
| | 1:10 | 0.0520 | 102.43 | 0.0170 | 65.35 |

$EC_{50}$ = the half-maximal effective concentrations; CTC = co-toxicity coefficient.

In general, fungicides show a specific antifungal mechanism, and the compounding application of different fungicides is an efficient strategy for the simultaneous control of different soil-borne diseases [34]. In addition, the utilization of synergistic combinations has also been shown to delay the development of disease resistance [35], reducing the number of chemical pesticides required [36].

Meanwhile, neonicotinoids are known to be highly effective against wheat aphids. In this study, we examined six neonicotinoids for their aphidicidal activity, namely, clothianidin, imidacloprid, nitenpyram, thiacloprid, acetamiprid, and thiamethoxam. All six neonicotinoids showed obvious aphidicidal activity; however, clothianidin was the most effective with an $LC_{50}$ of 0.3050 mg $L^{-1}$. Therefore, it was selected as the active aphidicidal ingredient in our seed coating agent (Table 2). An insecticide is an important component of seed coating agents, controlling aphid infestation and certain underground pests. According to a previous analysis of wheat cultivation, aphids are the most serious pest affecting wheat yield and quality Neonicotinoid insecticides are a class of novel pesticides and the main active component of many agricultural pest control measures [37]. Of the various types, clothianidin shows obvious contact activity, stomach action, and systemic activity, inducing a significant inhibitory effect on acetylcholine receptors [38]. Based on the above selection results, thifluzamide, fludioxonil, and clothianidin were used as active ingredients in our seed coating agent at an optimal ratio of 4:2:4 and a total content of 10%.

**Table 2.** Toxicity of each neonicotinoid insecticide against wheat aphid infestation based on the leaf-dip method after 48 h treatment.

| Neonicotinoid | Regression equation | $LC_{50}$ (mg·L$^{-1}$) | 95% CL (mg·L$^{-1}$) | $\chi^2$ |
|---|---|---|---|---|
| Clothianidin | 0.900 + 1.746x | 0.3050 | 0.231–0.390 | 8.016 |
| Imidacloprid | 0.344 + 1.819x | 0.6470 | 0.493–0.819 | 6.247 |
| Nitenpyram | 0.351 + 1.483x | 0.5800 | 0.411–0.768 | 2.195 |
| Thiacloprid | 0.057 + 1.773x | 0.9820 | 0.723–1.182 | 3.417 |
| Acetamiprid | 0.146 + 1.646x | 0.8160 | 0.620–1.052 | 4.067 |
| Thiamethoxam | 0.379 + 1.506x | 0.5600 | 0.396–0.740 | 3.802 |

$LC_{50}$ = lethal concentration to 50% of the population; CL = confidence limit.

The inactive components also form important components of seed coating agents, improving distribution across the seed surface [39]. These often include surfactants, film-forming agents, suspension concentrates, and pigments. In this study, we obtained an optimum formula by using the wet sand processing superfine grinding method. According to the orthogonal experimental design, 2-naphthalenesulfonic acid (NNO) was used as a wetting dispersant at a content of 1.70%. In addition, polyoxyethylene fatty acid (LAE-9, 2%) and emulsifiers (S-20, 2%) were used as nonionic surfactants, with magnesium aluminum silicate used as the thickening agent (1.70%) and ethylene glycol as the antifreeze agent (3%). The film-forming agent is also required to reduce the release of active ingredients. In a preliminary study, we developed a novel combination that had a relatively positive effect on the emergence rate, fresh weight, and root and stem lengths of emerging wheat seedlings, namely, polyvinyl alcohol and polyacrylamide + CMC at a content of 4%. Other additives included octanol (anti-foaming agent, 3%) and pigment red (dye agent, 6%) together with deionized water (69.15%) for a total volume of 100%. The detailed formulation of the resulting TFC seed coating agent is listed in Table 3, the main performance indexes of which were found to meet agent requirements.

**Table 3.** Main components of the novel seed coating agent (TFC).

| Component | Content % (g/g) | Property |
|---|---|---|
| Thifluzamide | 4.00 | Active ingredient |
| Fludioxonil | 2.00 | Active ingredient |
| Clothianidin | 4.00 | Active ingredient |
| NNO | 1.70 | Wetting dispersant |
| LAE-9 | 2.00 | Nonionic surfactant |
| S-20 | 2.00 | Emulsifier |
| XG | 0.15 | Thickener |
| Magnesium aluminum silicate | 1.70 | Thickening agent |
| Octanol | 0.30 | Anti-foaming agent |
| Ethylene glycol | 3.00 | Antifreeze agent |
| Pigment red | 6.00 | Dye agent |
| Polyacrylamide carboxymethyl cellulose | 4.00 | Film-forming agent |
| Deionized water | 69.15 | – |

### 3.2. Laboratory Tests

The effect of TFC on wheat germination and growth was investigated in the laboratory (Figure 1C–F). As shown, TFC had a positive effect on germination, plant height, and root length. At a ratio of 1:100 (agent: seed), the germinability and germination rates were 96.00 and 95.33% after 7 days, while seedling height and root length were 9.69 and 11.06, and fresh and dry weights were 3.26 and 0.54 g, respectively. Meanwhile, at a ratio of 1:500, a slightly better effect on germination and growth was observed, although the differences were not significant. The inhibitory rate was 99.68% at a ratio of 1:100 in roots inoculated with *R. cerealis* (Figure 1B), and the disease incidence improved rapidly within 20–26 days under all treatments. Moreover, the control efficiency on aphid infestations was greater than 95% (Figure 1A). Therefore, a TFC ratio of 1:100 was selected for the field trials.

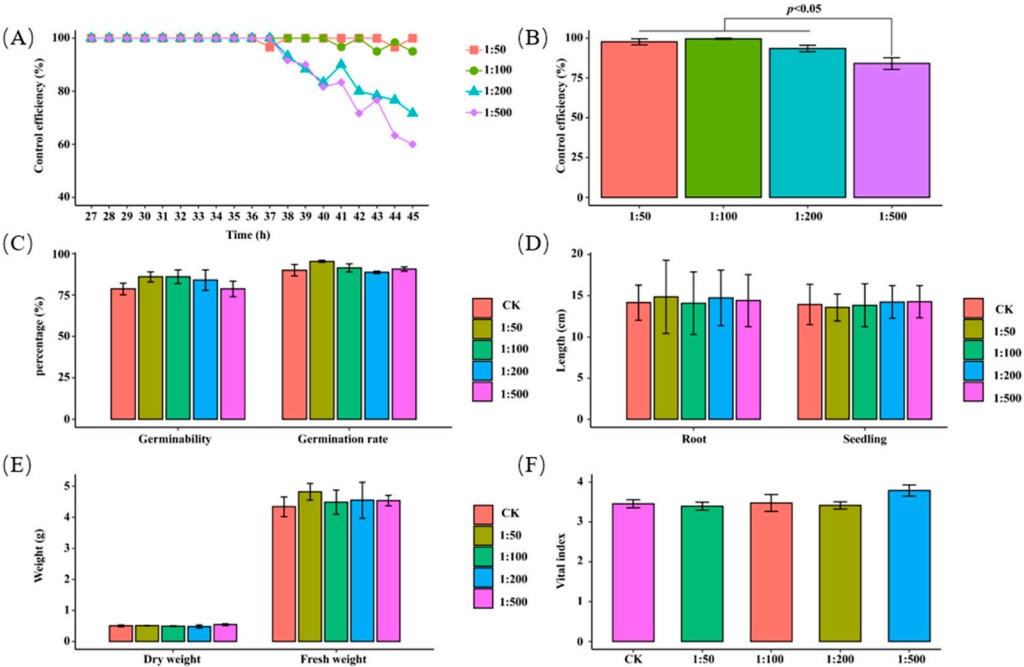

**Figure 1.** Effect of TFC in the laboratory. (**A**) Control effect on wheat aphid infestation (**B**) and wheat sharp eyespot. (**C**,**F**) Wheat germinability and germination rates, (**D**) seedling heights and root lengths, and (**E**) fresh and dry weights.

### 3.3. Field Tests

Field trials of TFC, DFT, and uncoated control samples were carried out in respective experimental plots. The effect on germination and growth and the control efficiency on soil-borne diseases and aphids are shown in Table 4. DFT is a traditional wheat seed-coating agent. Compared to DFT, TFC had obvious positive effects on wheat growth and was effective against wheat sharp eyespot and aphid infestation throughout the seedling stage. As shown in Table 4, TFC also promoted germination and growth compared to DFT, with an increase in the overall biomass. Seedling height and root length ranged from 31.4 to 32.3 and from 11.57 to 12.9 cm following treatment with DFT and TFC, while fresh and dry weights ranged from 63.19–75.60 to 25.37–26.54 g, respectively. Moreover, TFC effectively reduced the severity of wheat sharp disease and aphid infestation in the field, with control efficiencies of 86.7 and 96.7%, respectively.

**Table 4.** Effect of TFC and DFT in the field.

| Agent | Parameter | | | | Control Efficiency on Sharp Eyespot | Control Efficiency on Aphid Infestation |
|---|---|---|---|---|---|---|
| | Seedling Height (cm) | Root Length (cm) | Fresh Weight (g) | Dry Weight (g) | Control Effect (%) | Control Effect (%) |
| TFC | 32.27 ± 0.37 [a] | 12.9 ± 0.06 [a] | 75.60 ± 3.30 [a] | 26.54 ± 0.74 [a] | 86.7 [a] | 96.7 [a] |
| DFT | 31.40 ± 0.32 [ab] | 11.57 ± 0.43 [ab] | 63.19 ± 2.3 [a] | 25.37 ± 1.27 [a] | 81.3 [a] | 98.7 [a] |
| CK | 30.67 ± 0.26 [b] | 10.6 ± 0.55 [b] | 55.71 ± 1.28 [a] | 20.77 ± 1.32 [b] | - | - |

Means within a column followed by a different lowercase letter represent a significant difference ($p < 0.05$).

Considering the above growth parameters and control efficiencies, TFC appears to provide effective control of disease and pest infestation while having a positive effect on germination and growth in the field. These findings were consistent with the laboratory tests, confirming the validity of the experimental data.

### 3.4. Dynamic Changes in Pesticide Residues in the Soil and Plants

The analytical method was validated. The linear standard curves were obtained from the matrix-matched working standard solutions, at concentrations ranging from 0.005 to 1.0 mg L$^{-1}$. The concentrations of standard solutions were 0.5 mg/L, 0.2 mg/L, 0.1 mg/L, 0.05 mg/L, 0.02 mg/L, 0.01 mg/L, and 0.005 mg/L, respectively. A fortified study was carried out at levels of 0.01, 0.20, and 2.00 mg/kg to determine the recovery levels, precision, and limits of determination of the analytical method. These samples were processed as described above. The average recovery rates of thifluzamide in the soil, plant, and wheat grain samples ranged from 83.98 to 108.59% at the three fortification levels, which was within the acceptable range (70–120%) specified by the SANCO guidelines [40]. In addition, relative standard deviations (RSDs) were <10% for thifluzamide, indicating good repeatability. The mean recovery rates of fludioxonil and clothianidin ranged from 89.77–114.32% and 80.02–112.72%, respectively, with RSDs of < 6%, suggesting satisfactory accuracy. The LOQ of thifluzamide and clothianidin were 0.005 mg/L, while that of fludioxonil was 0.01 mg/L, which was within the concentrations required to achieve a signal-to-noise ratio (S/N) of between 3 and 10. In general, the recovery and precision results suggest that the method fulfilled the demands for analysis according to the residue quality control guide.

In our study, the residue levels of pesticides in soil differed. Thifluzamide had significantly higher soil residues and persisted longer than the other two pesticides. The dissipation results of thifluzamide, fludioxonil, and clothianidin in the soil are shown in Figure 2A. Initial deposits were 0.08, 0.06, and 0.05 mg kg$^{-1}$, with residues of all three pesticides showing an initial increase with a peak at 21 days followed by a subsequent decrease. This may have been caused by the binding of the residues within the soil after the seed coating treatment. Although there was an initial increase, the dissipation of fludioxonil was rapid, and the residues were undetectable at 54 days after treatment. Meanwhile, residues of thifluzamide and clothianidin in the soil were undetectable at 73 and 124 days after treatment, respectively. Although clothianidin shows higher control efficiency against wheat aphids, but most neonicotinoids persist in soils for a year or more and are water soluble, 80 to 98% of residues remaining in the soil of treated crops eventually move into surface waters or leach into groundwater, which led to significant reduction of macroinvertebrates in surface waters, and may also pose a risk to birds and rodents [20]. The selection of neonicotinoid insecticides used for seed treatment should be undertaken cautiously [41].

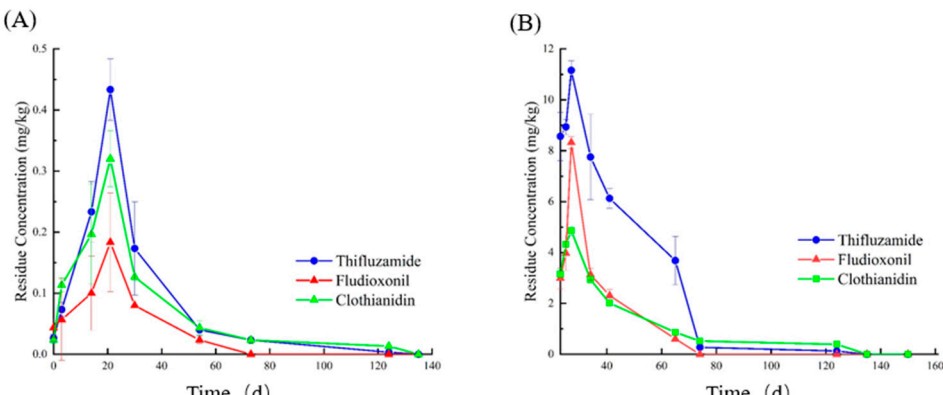

**Figure 2.** Dynamic changes in thifluzamide, fludioxonil, and clothianidin residues (mg kg$^{-1}$) in the (**A**) soil and (**B**) plant samples (the recommended dosage).

Figure 2B shows the dissipation data on thifluzamide, fludioxonil, and clothianidin in the plant samples. Each residue showed an initial increase followed by a gradual decrease, with peak values observed 4 days after treatment. Residues of thifluzamide remained at a significantly higher level than that of the remaining two pesticides until day

9, while those of fludioxonil decreased rapidly, reaching a low level by day 7. This rapid reduction in fludioxonil in the wheat plants is thought to be related to the low level of fludioxonil in the soil, while the low level of clothianidin in the wheat plants is thought to be related to water solubility, which is low in this insecticide at 0.304 g/L. In line with this, Stamm et al. found that soil moisture stress decreases the uptake of clothianidin [42], while Zhang et al. reported the effect of soil moisture on insecticide uptake and translocation [43]. In contrast, the residues of fludioxonil and clothianidin in plant samples remained at low levels throughout the sampling periods (Figure 2).

The final residues of thifluzamide, fludioxonil, and clothianidin in the wheat grain samples were detected after harvest, at levels of <0.05 mg/kg. China has an established maximum residue limit (MRL) of 0.5 mg/kg for thifluzamide in wheat (GB2763-2021). Here, the MRL of fludioxonil and clothianidin in the grains were 0.05 and 0.2 mg/kg, respectively, which was also within the limits of these safety guidelines.

*3.5. Effect of TFC on Earthworm Survival*

Earthworms (*Eisenia fetida*) are a critical bio-indicator of soil health [44,45]. The effects of pesticides on earthworms are significant in determining the bio-toxicity of pesticides in the soil [46,47]. Here, the number of earthworms under six concentrations of TFC was determined 7 and 14 d after treatment, as shown in Figure 3. Based on $LC_{50}$ values obtained on day 14, the results suggested that acute bio-toxicity was low, thereby conforming to the domestic standards on the safety of pesticide use. These findings further confirm that TFC is an environmentally-friendly alternative to more toxic seed coating agents.

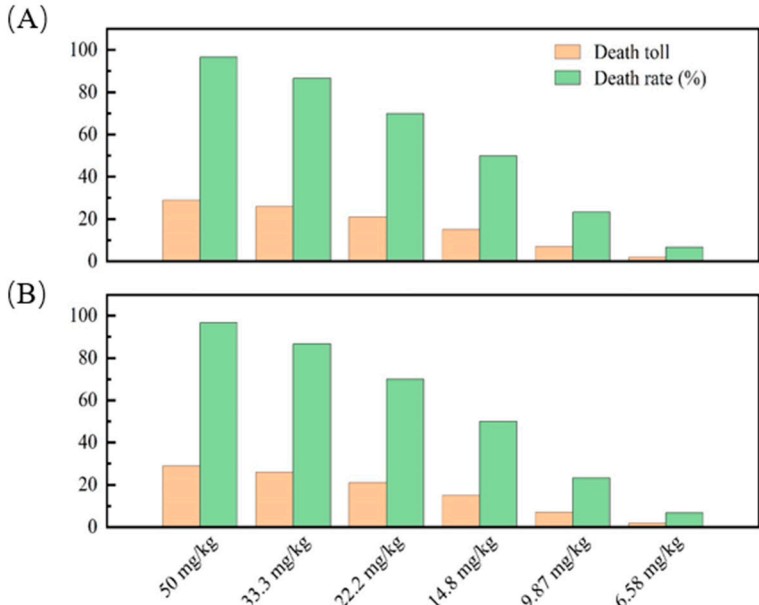

**Figure 3.** Earthworm mortality (**A**) 7 and (**B**) 14 days after treatment with six concentrations of TFC (mg/kg, dry soil): 6.58, 9.87, 14.8, 22.2, 33.3, and 50 mg/kg.

**4. Conclusions**

This study confirmed the successful preparation of an environmentally friendly wheat seed-coating agent (TFC) as an alternative to traditional, more toxic products. Compared with the conventional seed coating agent DFT, TFC had a positive effect on wheat sharp eye-spot and aphid infestation, as well as seed germination and growth. The results suggest that TFC can effectively control the incidence of major soil-borne diseases and pest infestations while improving the germination potential and rate of germination. The dissipation dynamics and quantities of thifluzamide, fludioxonil, and clothianidin residues in the wheat and soil were also studied using acetonitrile with UPLC-MS/MS. As a result, residues in the wheat grains at harvest were all below 0.05 mg/kg, both at the recommended dosage

and at 2.0 times the recommended dose. Furthermore, the artificial soil method used to test the biotoxicity of TFC in earthworms revealed low toxicity, at $LC_{50} > 10$ mg/kg. These results, therefore, provide an important foundation for the rational use of TFC in wheat production as an alternative to more toxic pesticides. Therefore, combining neonicotinoid seed treatments with fungicide seed treatments should be a suitable control measure for pest management at the wheat seedling stage. Moreover, the analytical method developed here to determine quantities of each pesticide in the soil and plants offers a rapid and reliable tool for scientific evaluation. The environmental risks from neonicotinoids in wheat fields need further research.

**Supplementary Materials:** The following supporting information can be downloaded at: https://www.mdpi.com/article/10.3390/agronomy12102330/s1, Figure S1. Flow chart showing the stepwise development of the seed coating agent. Table S1. Details of the test conditions for analysis of earthworm toxicity in the climate room. Table S2. MRM conditions for UPLC-MS/MS analysis.

**Author Contributions:** Conceptualization, H.C., C.C., X.R. and Q.G.; investigation, C.C., X.W., S.Y. and C.W.; formal analysis and writing (original draft preparation), C.C. All authors have read and agreed to the published version of the manuscript.

**Funding:** This research was funded by the Anhui Province Key Research and Development Project (Grant No. 202004a06020060).

**Data Availability Statement:** The datasets generated during and/or analyzed during the current study are available from the corresponding author upon reasonable request.

**Acknowledgments:** We gratefully acknowledge the Anhui Agricultural University for providing the experimental facilities. Additionally, we would like to acknowledge the staff of Anhui Agricultural University for their support and comments during the preparation of this manuscript.

**Conflicts of Interest:** The authors declare no conflict of interest.

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
