# Peer review of "Thifluzamide, Fludioxonil, and Clothianidin as Seed Treatment Can Efficiently Control Major Soil-Borne Diseases, Aphids (Aphidoidea spp.), and Residue Distribution in the Field"

_agronomy, doi:10.3390/agronomy12102330_

Round 1

Reviewer 1 Report

General Comments:

This manuscript describes experiments with a novel seed coating for wheat using a neonicotinoid and fungicides.  The manuscript is well written and covers many important aspects related to seed treatment such as optimum formulation, effect on germination, plant growth, efficacy against wheat sharp eyespot and aphid infestation, biotoxicity to earthworms and dissipation of residues in soil and  plant biomass.

The manuscript could be improved by giving more details about the dissipation field experiment and results.  I recommend also that the authors address the high risk of the neonicotinoid clothianidin for contamination through runoff events and its potential long-term risk to honey bees. 

More specific suggestions are given below.

Specific Suggestions:

pg. 2, line 52  Define DFT.

pg 4, line 157  Which of these dosages are shown in Figure 2A and 2B?

pg. 5, line 206  What are the components of the matrix matched solutions?  At what concentration?

pg. 6, line 267  I think the authors should address the fact that due to its low soil binding and high water solubility, clothianidin has a high risk for contamination through runoff events and therefore posses a long term toxic risk to honey bees.  Will this be an issue if used as a seed coating?

pg. 8, line 310   How is the control efficiency determined?

pg. 9, line 328  again, what and how did you prepare the matrix matched solutions?

pg, 9, line 343   Explain what you mean by initial deposits were 0.08, 0.06, 0.05 mg kg-1.  In the material and methods section on page 4, line 157, the dosages are listed as 225 and 450 g/hm2.  Please explain and clarify.

pg. 9, line 366  what dosage levels correspond to this Figure?

Reviewer 2 Report

The manuscript aims to present a seed treatment based on TFC, but within material and methods, the treatment is presented as a seed coating which is, in my opinion, a completely different technique and requires a different and deeper characterization method (physical and chemical). 

37-40

Seed-coating…toxicity.

Please explain more clearly how seed coatings can provide protection and at the same time reduce environmental toxicity. This claim seems to be in opposition to sentence at line 48-50 (however…health).

54-56

In this regard, … toxic products. How you can define as environmentally friendly a complex of 3 substances that are recognized as risky for the environment and non-target insects (some of them are banned in European countries)?  

56

What is the definition of seed coating and what is a ‘conventional’ coating agent DFT? Please provide some examples. Is TFC a filming agent? How does the coating form?

85

No explanation of the seed coating preparation method and no characterization of the coating has been provided in materials and methods. Have the seeds been soaked in the TFC preparation? How you can define it as a coating? Have you performed any tests to evaluate the formation of a shell on the surface of the seed (i.e. a coating)?

 102-105

Briefly, … room temperature.

Please, can you explain more clearly why it was employed DMSO for the preparation of the neonicotinoid-based solution? Why a leaf test was performed when you are researching about seed coating?

 118

‘Wheat seeds were coated with TFC then dried by airing for 20 min’. How was the coating made? How did you verify that the coating was really present on the seed surface? In my opinion, it is not possible to talk about a coating but just a poisoned seed. 

Author Response

Dear Editors and Reviewers:

Thank you for your letter and for the reviewers’ comments concerning our manuscript entitled “Thifluzamide, Fludioxonil, and Clothianidin as Seed Treatment can Efficiently Control Major Soil-Borne Diseases, Aphids (Aphidoidea spp.) and Residue Distribution in the Field” (ID:1902585). Those comments are all valuable and very helpful for revising and improving our paper, as well as the important guiding significance to our researches. We have studied comments carefully and have made correction which we hope meet with approval. Revised portion are marked in red in the paper. The main corrections in the paper and the responds to the reviewer’s comments are as flowing:

Answers to reviewers:

Reviewer #2:

  1. Comment: Line 37-40. Seed-coating…toxicity. Please explain more clearly how seed coatings can provide protection and at the same time reduce environmental toxicity. This claim seems to be in opposition to sentence at line 48-50 (however…health).

Response: Thank you for your useful suggestion. The seed coating agent is wrapped on the surface of the seeds and applied to the soil. Compared with foliar spray, it can effectively reduce the loss of liquid medicine and the toxicity to other organisms. We have revised it accordingly as following (You can see the related information in the line 37-41).

  1. Comment: Line 54-56. In this regard, … toxic products. How you can define as environmentally friendly a complex of 3 substances that are recognized as risky for the environment and non-target insects (some of them are banned in European countries)?

Response: Thank you for your useful suggestion. The results of the article showed that the residues of thifluzamide, fludioxonil and clothianidin in the soil and plants were all 0 at the late stage of wheat growth. There was no environmental risk. Clothianidin is banned in EU countries because of the high toxicity to bees when the agent is sprayed, and the seed treatment is applied to the ground without direct contact with bees, so there is no safety issue. Moreover, in my country, clothianidin is very common to be used for seed coating. (You can see the related information in the line 373-376, 386-388, 426-427).

  1. Comment: Line 56. What is the definition of seed coating and what is a ‘conventional’ coating agent DFT? Please provide some examples. Is TFC a filming agent? How does the coating form?

Response: Thank you for your useful suggestion. Seed coating is a technical measure to evenly coat the seed coating on the surface of the seeds. "DFT" (the 27% difenoconazole · fludioxonil · thiamethoxam, short for DFT) is the seed coating with the largest application area of wheat, purchased from Syngenta company. "TFC" (the 10% thifluzamide · fludioxonil · clothianidin, short for TFC) is a wheat seed coating agent developed by us, which contains a film-forming agent for film-forming. We have revised it accordingly as following (You can see the related information in the line 37-41, 60-63, 290-293).

  1. Comment: Line 85. No explanation of the seed coating preparation method and no characterization of the coating has been provided in materials and methods. Have the seeds been soaked in the TFC preparation? How you can define it as a coating? Have you performed any tests to evaluate the formation of a shell on the surface of the seed (i.e. a coating)?

Response: Thank you for your useful suggestion. We have revised it accordingly as following (You can see the related information in the line 115-132).

  1. Comment: Line 102-105. Briefly, …room temperature. Please, can you explain more clearly why it was employed DMSO for the preparation of the neonicotinoid-based solution? Why a leaf test was performed when you are researching about seed coating?

Response: Thank you for your useful suggestion. Room temperature is 25 degrees. DMSO is a universal solvent and has good solubility for neonicotinoids. And the dosage will not harm aphids. Wheat aphid determination method, one is a drop method, one is the immersion method. The drop method is mainly used to determine the effect of the contact on the insect. The pesticides we use are neonicotinoids, which are systemic, so we use the leaf-dip method.

  1. Comment: Line 118. ‘Wheat seeds were coated with TFC then dried by airing for 20 min’. How was the coating made? How did you verify that the coating was really present on the seed surface? In my opinion, it is not possible to talk about a coating but just a poisoned seed.

Response: Thank you for your useful suggestion. The coating of the seed coating depends on the film-forming agent, and the presence or absence of the coating can be effectively judged by the red dye. (You can see the related information in the line 292-293, 305-306).

In addition, we have entirely corrected all the mistakes and modified our manuscript. Once again, thank you very much for your constructive comments and suggestions which would help us both in English and in depth to improve the quality of the paper.

Kind regards,

Chao Chen

E-mail: chenchao05512022@163.com

Round 2

Reviewer 2 Report

The authors answered all the comments. 

The work can be accepted in this form after minor english revision